# An integrated atom array-nanophotonic chip platform with background-free imaging

Shankar G. Menon [1,5], Noah Glachman [1,5], Matteo Pompili[1], Alan Dibos [2,3,4] & Hannes Bernien [1] ✉

Arrays of neutral atoms trapped in optical tweezers have emerged as a leading platform for quantum information processing and quantum simulation due to their scalability, reconfigurable connectivity, and high-fidelity operations. Individual atoms are promising candidates for quantum networking due to their capability to emit indistinguishable photons that are entangled with their internal atomic states. Integrating atom arrays with photonic interfaces would enable distributed architectures in which nodes hosting many processing qubits could be efficiently linked together via the distribution of remote entanglement. However, many atom array techniques cease to work in close proximity to photonic interfaces, with atom detection via standard fluorescence imaging presenting a major challenge due to scattering from nearby photonic devices. Here, we demonstrate an architecture that combines atom arrays with up to 64 optical tweezers and a millimeter-scale photonic chip hosting more than 100 nanophotonic cavities. We achieve high-fidelity ( ~ 99.2%), background-free imaging in close proximity to nanofabricated cavities using a multichromatic excitation and detection scheme. The atoms can be imaged while trapped a few hundred nanometers above the dielectric surface, which we verify using Stark shift measurements of the modified trapping potential. Finally, we rearrange atoms into defect-free arrays and load them simultaneously onto the same or multiple devices.

Neutral atom arrays trapped in optical tweezers show promise as quantum information processors due to their characteristic scalability[1,2] and programmability with any-to-any connectivity[3,4]. Recently, high-fidelity entangling operations[5–7] and the capability to perform mid-circuit readout and feedback[8–11] have also been demonstrated in atom-array systems. Combining these systems with a photonic interface can further enhance their capabilities by enabling quantum communication[12], blind and distributed quantum computation[13–16], mid-circuit and faster readout techniques[17], distributed sensing[18,19], as well as by extending the set of long-range interaction Hamiltonians that can be simulated[20–26].

Trapped atoms have been integrated with macroscopic Fabry-Perot cavities[27–30], fiber cavities[31], nanofiber-based waveguides[32–37], as well as nanophotonic waveguides and cavities[38–40]. Among these, nanophotonic cavities and waveguides offer some of the highest atom-photon interaction strengths owing to their small mode volumes and high-quality factors[41]. Up to two atoms have been deterministically trapped on top of a nanophotonic cavity and entangled using cavity-carving techniques[41,42].

Scaling systems that combine atoms with photonic interfaces and incorporating atom array capabilities is not a straightforward task. The presence of a dielectric structure can significantly impact

[1]Pritzker School of Molecular Engineering, University of Chicago, Chicago, IL 60637, USA. [2]Argonne National Laboratory, Center for Nanoscale Materials, Lemont, IL 60439, USA. [3]Nanoscience and Technology Division, Argonne National Laboratory, Lemont, IL 60439, USA. [4]Argonne National Laboratory, Center for Molecular Engineering, Lemont, IL 60439, USA. [5]These authors contributed equally: Shankar G. Menon, Noah Glachman. ✉e-mail: bernien@uchicago.edu

the process of loading atoms into traps. While atoms have been cooled and loaded directly into traps in the vicinity of a single nanophotonic cavity[40] or waveguide[32–37], the viability of directly loading traps near a chip-scale structure has been undermined by unbalanced scattering and reflections from the dielectric surface. Techniques such as loading from an atomic fountain[43,44] or a double magneto-optical trap (MOT)[38] have been implemented to overcome this challenge near larger dielectric structures. Additionally, methods such as stage-based transport of a tweezer array, an optical conveyor belt, and an optical funnel formed through a transparent dielectric structure have been employed to increase the trap-filling fraction near chip-scale structures[45–49].

Fluorescence imaging of atoms in the vicinity and on top of nanophotonic devices presents another key challenge due to the scattering of the imaging beams from the device. Transmission and reflection measurements have been explored to estimate the number of atoms coupled to a device[32–38,43,44], enabling fast readout. However, these methods only provide global system information rather than site-resolved atomic state information. To achieve the site-resolved, single-shot readout necessary for quantum information processing and networking, methods such as confocal microscopy[41], exciting atoms through propagating waveguide modes[50], and polarization filtering combined with spatial filtering[48] have been implemented. However, these techniques have been limited to either a few atoms, single devices, or specific device geometries. Demonstrating atom array techniques such as rearrangement and single-shot readout of large atom arrays near arbitrary nanophotonic devices remains an outstanding challenge.

In this work, we introduce an experimental platform combining an atom array with a silicon nitride on a silicon chip hosting more than 100 nanophotonic devices along with a background-free imaging scheme to overcome scattering from the nearby chip. A semi-open chip geometry, where devices are suspended from the edge of the chip, provides sufficient laser cooling access to enable MOT formation near the chip structure. A free-space atom array can be loaded in the open space to the side of the chip. We find that the presence of the chip has minimal effect on the atom loading characteristics into the tweezers with loading probabilities and temperatures similar to conventional atom array experiments[51]. Further, light can be coupled in or out of the nanophotonic devices via efficient free-space coupling, enabling fiber-free photon coupling to any devices inside the chamber[45,52].

A multichromatic imaging technique[53–56] suppresses the device scattering, enabling single-shot readout of the entire atom array close to, or even on top of, the devices using an electron-multiplying charge-coupled device (EMCCD) camera similar to the standard readout method for free-space atom arrays. Combining the above capabilities, we demonstrate that we can rearrange atoms and load them onto multiple devices at the same time or to a single device, where they can be imaged in a single shot with our imaging technique.

The techniques and methods presented here represent a general recipe for integrating arrays of atoms with a wide range of nanophotonic structures, including alligator waveguides[38], corrugated cavities[57], or a more complex combination of 1D cavities[58] where the key ingredients are a semi-open chip geometry, multichromatic fluorescence imaging, and a method of coupling light in and out from the nanophotonics. This platform combines the measurement and rearrangement capabilities of atom arrays with the ability to engineer the photonic environment via integrated cavities and waveguides, representing an enabling step towards multiplexed telecom quantum networking with resonant cavities[59,60], fault-tolerant distributed quantum computing with Rydberg integration[61], and demonstrations of novel many-body phenomena in atom-waveguide systems including self-organization of atoms and the generation of arbitrary photonic states[22,24].

## Results

Our platform consists of an array of optical tweezers formed by a pair of crossed acousto-optic deflectors (AODs) that can be used to trap, move, and rearrange individual atoms in our ultra-high vacuum chamber as depicted in Fig. 1a. We load our tweezers with cesium atoms $\lesssim 100\,\mu m$ away from a $600\,\mu m$ thick $2\,mm \times 8\,mm$ chip hosting over one hundred nanophotonic crystal cavities ~ $60\,\mu m$ long (Fig. 1a inset), though the techniques and results presented here are independent of the device type and broadly apply to any dielectric structures. The devices are fabricated from 330 nm thick silicon nitride on top of a silicon substrate, which is completely undercut from the device region, enabling the tweezer beams to pass by the chip even in close proximity to the devices (for details, see supplementary information). The nanophotonic devices are characterized via a coupling lens outside the chamber which produces a diffraction-limited spot that is mode-matched to the end mode of the tapered waveguides appended to our nanophotonic cavities (see Fig. 1a, b)[45,52]. The coupling efficiency is currently measured to be 20%, and simulations show that up to 91% is, in principle, possible (see supplementary information). We cool the atoms by forming a MOT and load atoms into the optical tweezers during polarization gradient cooling (PGC) in a loading region tens of microns from the devices (see Fig. 2b). The optical beams that form the MOT and image the atoms are partially reflected by the chip and devices, rendering the standard fluorescence imaging technique of driving the 852 nm cycling (D2) transition impractical as the scattering from the surface of the solid structure is many orders of magnitude more intense than the atomic fluorescence. We demonstrate the scale of this problem in Fig. 2a, where we show that even away from the devices, in our loading region, this scattering leads to signals on the order of hundreds to thousands of photons per pixel, compared to our atomic signals, which are typically on the order of ~1.6 photons per pixel (25 photons detected within a $4 \times 4$ pixel region of interest). On the region of the devices, we have several pixels detecting more than 350,000 photons. At higher EM gain settings necessary for imaging single atoms, signals of this magnitude would rapidly damage the camera sensor.

We use a background-free imaging technique to circumvent this scattering issue where we drive a two-photon transition to an excited state (here the $7S_{1/2}$ state), which has two decay pathways (see Fig. 1c). We spectrally filter out the excitation wavelengths and image the fluorescence from the atoms along the other decay path at 895 nm. With this method, we filter out all of the unwanted backgrounds due to the scattering from the devices and achieve high detection fidelities (see Fig. 2c). We show an averaged fluorescence image of our array both away from the devices in the loading region (Fig. 2b inset) and interleaved between the devices (Fig. 2b). A typical histogram of the signal received from a single site (from Fig. 2b inset) showing the ability to discriminate with high fidelity between the presence and absence of an atom is shown in Fig. 2c. We estimate an imaging fidelity of 99.2% by fitting the bimodal histogram and examining the overlap of the signal and the background, similar to typical fidelities obtained via D2 line fluorescence imaging[8]. Here, we use a 40 ms exposure time, also comparable to the timescale of standard fluorescence imaging[51], making this method applicable to a wide range of experimental setups that suffer from background light.

In Fig. 2d, we plot the magnitude of the fluorescence signal detected from the atoms as we vary the detunings of the two drive lasers. The 852 nm laser detuning on the $x$-axis is measured with respect to the bare $6S_{1/2}\ F = 4 \rightarrow 6P_{3/2}\ F' = 5$ transition and the 1470 nm laser detuning on the $y$-axis is measured with respect to the bare $6P_{3/2}\ F'=5 \rightarrow 7S_{1/2}\ F'' = 4$ transition. The detected signal falls off as the 852 nm detuning approaches the $6S_{1/2}\ F = 4 \rightarrow 6P_{3/2}\ F' = 5$ transition on the right side of the plot and when it approaches the $6S_{1/2}\ F = 4 \rightarrow 6P_{3/2}\ F' = 4$ transition ($-251\,MHz$ on the $x$-axis) on the left side of the plot. We

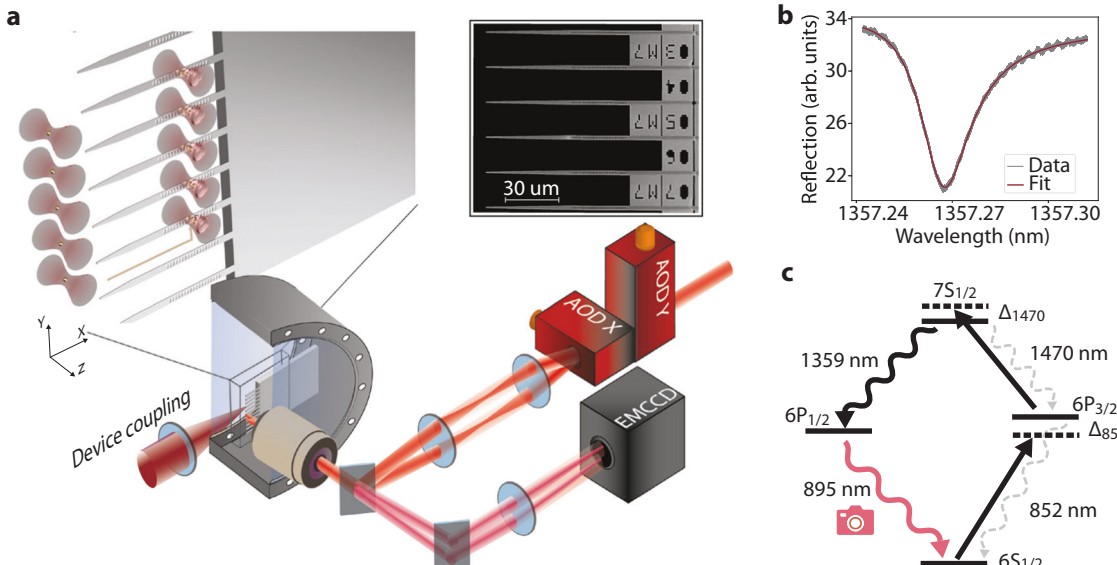

**Fig. 1 | The atom array–nanophotonic chip platform. a** Schematic of the platform depicting our optical tweezer array manipulating single cesium atoms near a chip hosting an array of nanophotonic devices. The zoomed-out view shows the chip inside a stainless steel ultra-high vacuum chamber along with the key optical components used to control and image the system (AOD: acousto-optic deflector, EMCCD: electron-multiplying charge-coupled device). The inset shows a scanning electron microscope (SEM) image of the nanophotonic devices at the edge of our chip. **b** Reflection spectra of a nanophotonic cavity inside the chamber measured using the free-space coupling technique. The cavity is designed for our

future quantum networking experiments, however, the techniques presented here are general and applicable to cavities, waveguides, or other photonic structures. **c** Level structure that illustrates the background-free imaging technique. We doubly excite the atoms from the $6S_{1/2}$ ground state following the straight black arrows through the $6P_{3/2}$ intermediate state and spectrally filter the fluorescence reaching the camera to only image the 895 nm decay path from the $6P_{1/2}$ level shown in red. Here, $\Delta_{852}$ is the detuning from the bare $6S_{1/2}$ $F = 4 \rightarrow 6P_{3/2}$ $F' = 5$ transition, and $\Delta_{1470}$ is the detuning from the bare $6P_{3/2}$ $F'=5 \rightarrow 7S_{1/2}$ $F''=4$ transition.

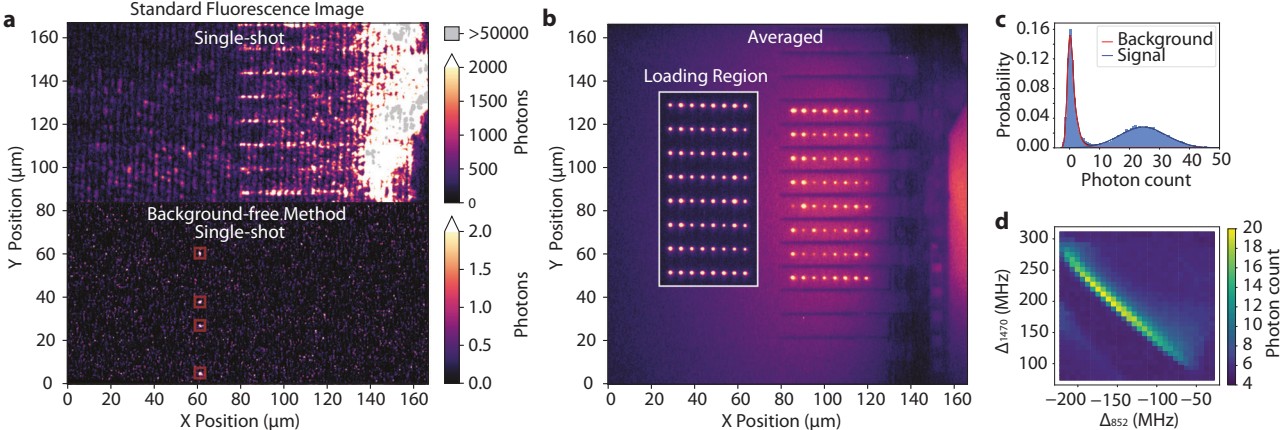

**Fig. 2 | Background-free imaging of the atom array. a** Top: Fluorescence image using the standard 852 nm (D2) cycling transition near our nanophotonic chip. Despite lowering the electron multiplying gain (EM gain) process on our camera to 10 from our typical value of 1000, the image is still saturated at the single atom scale, even tens of microns away from the devices. Bottom: A single-shot image of atoms (inside the red boxes) taken using the background-free technique developed in this work. Here, both images are taken with a 40 ms exposure time. **b** An averaged fluorescence image of the atom array interleaved between the nanophotonic device array. No post-processing has been applied here beyond averaging the individual images. A small residual background makes the devices appear as dark

shadows. We attribute this background to fluorescence at our imaging wavelength from the silicon base layer of the nanophotonic chip. The inset shows an averaged fluorescence image of an 8 × 8 atom array in the loading region. **c** A typical histogram of the detected 895 nm photons within a 4 × 4 pixel region of interest over a 40 ms exposure time in the loading region. The bimodal distribution allows us to distinguish between the presence and absence of an atom with high fidelity ≥ 99.2%. **d** Average magnitude of the detected fluorescence signal from the atoms as a function of the two drive laser detunings. The signal falls off as the 852 nm laser approaches hyperfine resonances on each side of the plot, leading to atom loss from the tweezers.

attribute this to atom loss from the tweezer due to resonant heating. The optimal point is far from a ground state resonance, where cycling on the lower transition is suppressed, and the two-photon excitation to the doubly excited state is the dominant process. The overall feature is blue-shifted from the bare two-photon resonance by tens of MHz due to the AC Stark shift from the tweezer.

Another crucial capability required for nanophotonic integration is the ability to move atoms onto the photonic chip in close enough proximity to couple the atoms to the light field of the devices. We achieve this by adiabatically translating the optical tweezers that are loaded with individual atoms from the loading region until they aim directly at the devices, where the tweezer beams partially reflect

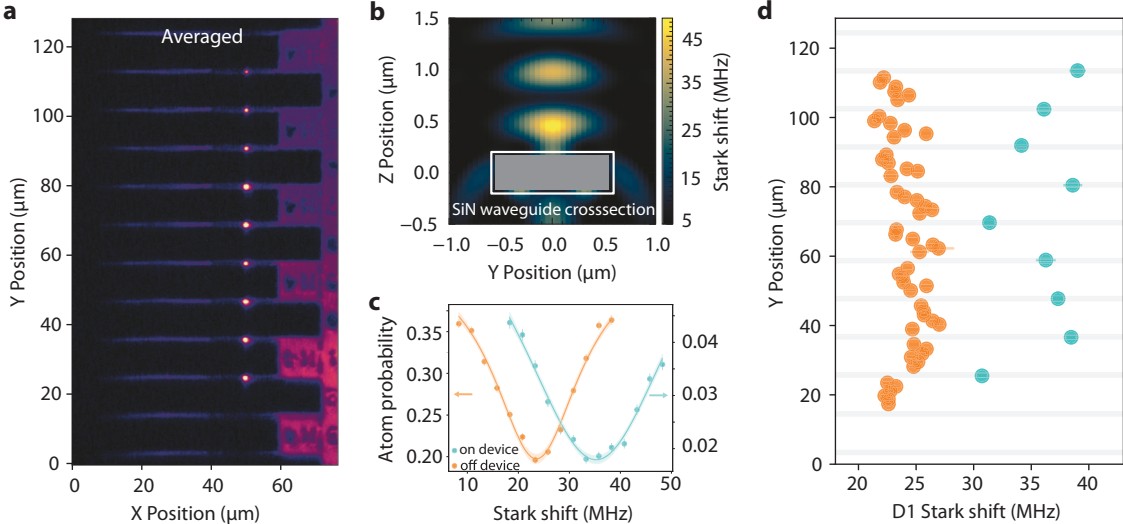

**Fig. 3 | Imaging atoms on top of devices. a** The averaged fluorescence from 15,000 individual images of atoms loaded onto the nanophotonic devices overlaid with an image of the devices (see the "Methods" section for image processing details). **b** Cross section of the nanophotonic device and the expected Stark shift on the D1 transition for the different intensity maxima formed on top of the device by the partially reflected tweezer. **c** Averaged Stark shift measurements on the 895 nm $6S_{1/2}\,F = 3 \to 6P_{1/2}\,F' = 4$ transition in between the devices (orange) and on top of the devices (cyan). Centers estimated from Lorentzian fits to similar plots at each

position are used to generate (**d**). Error bars represent the standard error of the mean. **d** The centers of the Stark shift curves for individual atoms as a function of their positions. The gray lines are estimated device positions from (**a**). The Stark shifts are larger in magnitude when the atoms are on top of the devices, as expected from the modified trapping potential on top of the devices as shown in (**b**) (for the trapping potential away from the devices, see supplementary information). The error bars represent the standard error of the Lorentzian fit.

backwards, forming standing wave traps above the devices as shown in Fig. 3b. We show an averaged image of the atoms loaded on top of the devices overlaid with an image of the devices (see the "Methods" section for image processing details) in Fig. 3a. The goal is to load the atoms into the closest intensity maximum ~300 nm from the surface of the silicon nitride, where they can strongly couple to the cavity mode. The distance between the closest intensity maximum and the surface is set by a combination of the tweezer wavelength and the thickness of the device. This standing wave trap has an intensity ~2 × the intensity of the tweezer in free space. In order to determine whether we are loading the atom to the standing wave trap, we probe the AC Stark shift experienced by the atom, which is proportional to the intensity of the trap, on the $6S_{1/2}\,F = 3 \to 6P_{1/2}\,F' = 4$ (D1) transition at 895 nm and compare it to the Stark shift the atom experiences in the free-space tweezer. We probe this specific transition because our tweezer wavelength of $\lambda = 935$ nm is magic for the $6S_{1/2} \to 6P_{3/2}$ transition, meaning that the Stark shifts induced on the $6S_{1/2}$ and $6P_{3/2}$ levels are equal in both sign and magnitude, rendering it insensitive to the intensity variation between the trapping potentials.

We begin this Stark shift measurement by loading the tweezers with atoms in the loading region, moving the tweezers between the devices, and then moving the tweezers onto the devices from the side. We then apply a variable frequency 895 nm laser pulse in order to blow out the atoms from the tweezer when the pulse is resonant with the Stark-shifted atomic transition. Finally, we image the atoms to detect the survival rate. In Fig. 3c, we show typical blow-out survival curves showing the increased Stark shift when the atoms are loaded onto the devices. In Fig. 3d, we plot the fitted centers of the blow-out survival curves as we move the atoms across the device region, showing that the observed increase in the Stark shift only occurs when atoms are directly on top of the devices where they are trapped in the higher intensity standing wave potential shown in Fig. 3b. From the observed Stark shifts and blow-out curves of the individual atoms, our modeling indicates varying loading probabilities across the devices with a maximum loading probability of 29% into the first intensity maximum (see supplementary information for details). The tweezer power, aberrations, and the angle between the devices and the tweezer focal plane contribute to the

variations in the observed Stark shifts and loading probabilities across the devices and can be further optimized in future experiments.

In order to fully integrate atom arrays with nanophotonics, we must be able to rearrange the atoms into defect-free arrays and then load them onto the devices. To achieve this, we start by taking an image of the initial random loading (25 ms) and rapidly process (~7 μs) the image data into an occupation matrix. This information is then used to drop the unoccupied tweezers and compress the remaining atoms into a defect-free array before translating the whole array to the devices (see the "Methods" section for details). The experimental sequence for this procedure is shown in Fig. 4a.

The final step of this procedure can be done either by loading one atom per device or by loading multiple atoms to a single device. We demonstrate both capabilities in Fig. 4 by showing initial images of the randomly loaded atoms in free-space tweezers next to a second image of the same atoms post-rearrangement loaded onto the devices in each configuration, one atom per device (Fig. 4b) and three atoms on a single device (Fig. 4c). We also show averaged post-rearrangement images for 15,000 stochastic loading and rearranging events in each configuration, where seven to eight atoms can be seen in each of the post-rearrangement averaged images from the nine initial loading sites.

## Discussion

We have realized a platform for combining the capabilities of atom arrays with an integrated chip capable of hosting many nanophotonic devices. We load large arrays of atoms near the nanophotonic chip and image them with fidelities greater than 99.2% with the background-free imaging technique. This capability enables us to load and image atoms on top of the nanophotonic cavities and extract information about the Stark shift to demonstrate atom loading onto the first few intensity maxima on top of the devices. With further optimizations of atomic temperature, tweezer parameters during loading, and device parameters, atoms can be deterministically loaded to the intensity maximum of interest[40]. The rate of deterministic placement of atoms in Fig. 4 is currently limited by atom survival through the first image, probability of successfully loading to the device, and survival during imaging on top of the devices. A small angle between the device plane

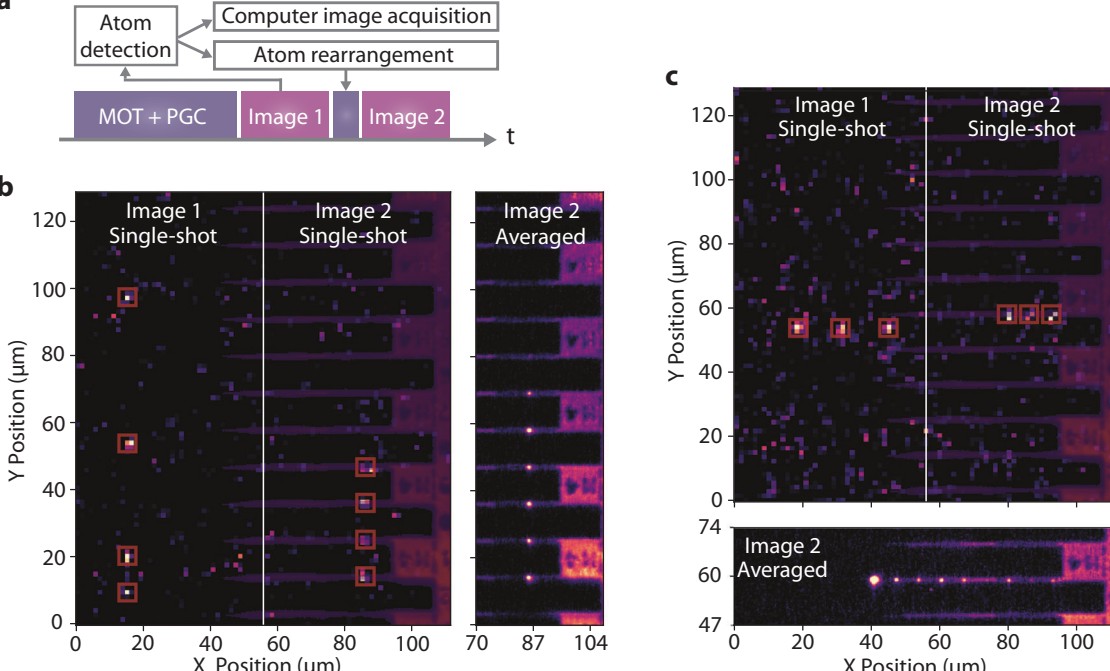

**Fig. 4 | Defect-free atomic rearrangement and loading onto devices.**
**a** Experimental sequence used for defect-free rearrangement of the atoms from free-space onto the devices. After cooling the atoms and loading them into the tweezers, we take Image 1 to detect the stochastic loading pattern and use this information to rearrange the array into a defect-free configuration which we then translate over and onto the devices before we take Image 2 (see the "Methods" section for details). **b** Image 1 shows a single-shot image of the randomly loaded atoms in a nine tweezer array in the loading region. After detecting these atoms in the first image, they are rearranged into a defect-free array with the same spacing as

the devices. This compressed array is then loaded onto the devices and the final configuration is shown in Image 2. An averaged image of the final configuration after this procedure is shown to the right (see methods for image processing details). For all images in this figure, a 25 ms exposure time was used to increase the atomic survival rate through the first image. **c** Here, an array of stochastically loaded atoms in Image 1 are rearranged and loaded onto a single device as shown in Image 2. In the bottom plot, the averaged image of the final configuration is shown. The bright single atom on the left of the averaged image is an atom that is rearranged to a position outside the device region.

and the tweezer plane also causes variation in the loading distribution to the various intensity maxima across the devices as this distribution is highly sensitive to the relative $Z$ position of the device and the tweezer focus. Improved device alignment and techniques such as stroboscopic imaging or Raman-sideband cooling of the atoms can be used to further improve these metrics.

The devices presented in this work are nanophotonic cavities embedded in waveguides. Currently, the cavities are not resonant with the atomic transition, preventing direct atom–cavity interactions. In future work, tighter fabrication tolerances and thermal tuning can be incorporated to bring devices on resonance and to stabilize them to the atomic transition[62]. The atom trapping distance of ~300 nm measured in this work is expected to give an atom–cavity interaction strength in the range of $2\pi \times 600$ MHz[59], similar to other 1D photonic cavities[41]. These interaction strengths are two order of magnitude larger than what is typically observed in mirror cavities[63] and an order of magnitude larger than fiber cavities[31]. Along with increased interaction strengths, photonic cavities also have the highest photon loss rates from the cavities, making them suitable for fast photon extraction. In future work, these cavities can be utilized to entangle a subsection of the atoms with photons enabling multiplexed entanglement generation[59,60]. Furthermore, with the inherent rearrangement capability in our system, atoms can be moved sufficiently far away from the device for Rydberg-mediated gates[64]. Incorporating these capabilities would readily enable quantum simulation and computation operations while preserving the ability to distribute remote entanglement, providing a path towards multiplexed quantum repeaters and fault-tolerant distributed quantum computation[61].

Furthermore, our semi-open chip geometry is flexible and can be tailored towards many potential applications and a wide variety of

nanophotonic structures. Alternative methods of coupling light to the nanophotonics such as grating couplers or tapered optical fibers via the chip surface could be explored for possible applications where the ejected mode cannot overlap the free space array. Overhanging nanobeams can be extended to loading tracks fabricated on the chip surface to provide atomic access to more complicated nanophotonic devices and circuits that cannot overhang the chip edge[65]. Waveguides for quantum simulation and cavities for atom-photon entanglement can be incorporated into the same chip for distributed quantum simulation. Currently, the majority of the chip surface is not utilized, enabling further opportunities such as the integration of beam splitters, modulators, and detectors directly on the chip.

## Methods
### Platform details
Our experiment starts with ~170 ms of MOT loading to maintain a 1:1.5 MOT on/MOT off ratio in our experiment cycles (see supplementary information). The atoms are sourced from a heated dispenser in the same chamber located ~5 cm from the MOT position. This is followed by 10 ms of polarization gradient cooling (PGC) during which atoms are stochastically loaded into the tweezers. We observe 55% loading efficiency into the tweezer array and atomic temperatures of approximately 50 μK. The six beam MOT is formed by two retroreflected MOT beams in the plane of the chip, a MOT beam sent through the tweezer objective, and a counter propagating MOT beam sent though another objective located on the opposite side of the chamber. A home-built ECDL laser at 935 nm and amplified using a tapered amplifier (MOGLabs MOA) is used to form the optical tweezer array. The RF frequencies that drive the AODs (AA Opto-Electronic) are generated using an AWG for 8 × 8 array of tweezers and an FPGA for

1 × 9 and 9 × 1 arrays (Quantum Machines, OPX for both methods). A custom, high NA objective (Special Optics 0.6 NA) is used to focus the tweezers into the vacuum chamber. Atomic fluorescence at the imaging wavelength is collected using the same objective and imaged onto an EMCCD camera (NüVü HNü 512 Gamma). For rearrangement, we use fast detection and feedback directly to the OPX using an intermediate module that processes the camera images into occupation matrices and communicates that information to the OPX in real time (Observe camera readout module, Quantum Machines).

## Device fabrication

The nanophotonic chip is made from a 340 nm-thick layer of commercially available stoichiometric LPCVD-grown silicon nitride ($Si_3N_4$) on top of a 600 μm-thick silicon substrate (Silicon Valley Microelectronics). The nanophotonic devices are 1.1 μm wide and 63 μm long and are repeated every 11 μm. The cavity design, electron-beam lithography (EBL), and reactive-ion etching (RIE) steps are carried out at the Center for Nanoscale Materials at Argonne National Laboratory, and the remaining processes and design are carried out in the Pritzker Nanofabrication Facility at the University of Chicago. The details of the fabrication are provided in the supplementary section.

## Image processing

For the single-shot images labeled Image 1 and Image 2 in Fig. 4b and c, we first plot the two single-shot atom images and then plot a semi-transparent image of the devices over them to show the device locations without obscuring the raw atom image data underneath. For Figs. 3a, 4b Image 2 Average, and Fig. 4c Image 2 Average, background subtracted atom image is scaled and added to the device image. A small offset is added before scaling to avoid any negative signals. This sum then forms the final plot shown.

## Data availability

The data that support the findings of this study are available from the corresponding author upon request.

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

## Acknowledgements

We thank Kevin Singh, Yu-Hao Deng, Yuzhou Chai, Dahlia Ghoshal, Haley Nguyen, and Harry Levine for insightful discussions. We also thank Ramon Szmuk and Alex Kotikov for their help integrating the Observe and the OPX into our platform. We gratefully acknowledge funding from the NSF QLCI for Hybrid Quantum Architectures and Networks (NSF award 2016136), the NSF Quantum Interconnects Challenge for Transformational Advances in Quantum Systems (NSF award 2138068), the NSF Career program (NSF award 2238860), and the Sloan Foundation. Photonic crystal cavity simulation, design, and device fabrication were performed at the Center for Nanoscale Materials, a U.S. Department of Energy Office of Science User Facility, supported by the U.S. DOE, Office of Basic Energy Sciences, under Contract No. DE-AC02-06CH11357.

## Author contributions

S.G.M., N.G., M.P., and H.B. contributed to the experiments. S.G.M. and A.D. fabricated the nanophotonic devices. S.G.M., N.G., M.P., and H.B. contributed to the analysis of the results and preparation of the manuscript.

## Competing interests

S.G.M., N.G., and H.B. have included the work described in this article in a patent application filed with the US Patent and Trade Office by the University of Chicago. The remaining authors declare no competing interests.
