## [Peer Review File · Nature Communications]

An integrated atom array-nanophotonic chip platform with
background-free imagingREVIEWER COMMENTS

Reviewer #1 (Remarks to the Author):

Coupling neutral atoms trapped in optical tweezers with nanophotonic devices features broad applications in quantum information processing and quantum simulation. One crucial technical difficulty in its realization is that standard fluorescence imaging will fail due to strong scattering near the photonic devices. In this article, the authors presented a background-free imaging method by driving a two photon transition to an excited state and imaging one of the decay pathways. With the ability to perform high fidelity imaging, the authors further demonstrated the rearrangement of stochastic loaded optical tweezers into a (small) defect free array, and moving the atoms over the nanophotonic devices. The high fidelity imaging through the background free scheme is the main technical achievement of this paper. The rearrangement and transport of atoms is currently limited to small array sizes (3 - 4 atoms), but the authors are upfront about this and outlines directions for future improvements. In our view there are sufficient technical advances demonstrated here to justify publication in Nature Communications. The manuscript is very clearly written. The data is rigorously analyzed and well-presented. We have a few comments and questions for the authors.

1. The paper uses a mix of single shot images and averaged images. These are clearly distinguished in the captions and the main text, but it'll be easier for the reader if they are labeled directly on the images.
2. The authors report an imaging fidelity of 99.2%. This number should be put into context with state of the art imaging fidelity for free space optical tweezers, as well as those coupled to photonic crystals.
3. The authors mention that the standing wave trap has $\sim 2x$ the intensity as a free space tweezer. I would naively expect the field to be $2x$ higher and intensity to be $4x$ higher. Could the authors explain this?
4. The term "magic" (wavelength) is used without explanation. The concept may not be familiar to all readers.
5. Legends in Fig 3c are too small.
6. In image 2 of Fig. 4b, the individual atoms are a bit hard to see by eye. Consider

highlighting their locations.

Reviewer #2 (Remarks to the Author):

The manuscript by Menon*, Glachman* et al reports on an experimental platform that seeks to couple atom arrays to nanophotonic chips. Specifically, they address two challenges: (1) loading atoms near chip-scale structures while retaining the reconfigurability and control of free-space tweezers and (2) imaging the atoms with high fidelity when they are in the vicinity of the chip structures. They tackle the first challenge by using tweezer arrays to stochastically load atoms from a MOT, rearranging the tweezers to achieve unity filling, and moving the tweezers to the chips using radiofrequency control of the AODs. They tackle the second challenge by spectrally removing the exciting wavelengths and filtering only the 895nm fluorescence, achieving background-free imaging. Integrating individual atoms with nanophotonic structures is an active field of research with implications for networked quantum systems for quantum computation and communication.

The paper is well-written and concise, and the claims are well supported by evidence. Prior work is appropriately acknowledged, and the methods are described in sufficient detail to ensure reproducibility of the results.

While I believe the technical advances presented here are significant for the field, the scale and novelty of the advances need to be justified in the context of other works which have addressed and overcome similar challenges of atom loading (refs 43-49) and imaging (refs 41,48,50) near surfaces.

Below are questions that need to be clarified before I can recommend publication in Nature Communications.

Q1) Claims on generality of chip geometry: In page 1, the authors claim previous attempts have “been limited to... specific device geometries” and imply that their work addresses the outstanding challenge of “readout of large atoms arrays near arbitrary nanophotonic devices”. Later, they claim “the techniques... represent a general recipe”.

While the semi-open ship design is indeed very clever, I don't see how such claims of generality are justified when the presented designs are simply well-established 1D cavity waveguides. How would arbitrary 2D nanophotonic devices be integrated with atom arrays with the presented design? In the Discussion, the authors briefly mention extensions to "loading tracks fabricated on the chip" which could potentially lead to general recipes with arbitrary structures, but without providing further clarity on how such an extension is feasible, one might as well say that their design simply represents another "specific device geometry".

Q2) How do the authors propose to work with multiple nanophotonic devices (eg., scenario in 4b) where light would have to be coupled with good mode-matching into the tapered ends of each device? The present work only demonstrates coupling to a single device using a focussed free-space beam from outside the chamber.

Q3) In the Discussion, the authors say "we load large arrays of atoms near the nanophotonic chip", which seems unwarranted given that atom array experiments these days (including work by this group) routinely work with hundreds of atoms. The atom numbers presented here ($2 < n < 10$) are comparable to similar works of atoms near nanophotonic surfaces (refs 45, 48).

Q4) Novelty of advances: The authors cite previous works that have tackled the challenges of loading (refs 43-49) and imaging (refs 41,48,50) atoms near nanophotonic surfaces. Following this, they state "However, these techniques have been limited to either a few atoms, single devices, or specific device geometries", implying that their overcoming of all these challenges in one platform is the novelty. I would argue that the results in manuscript also demonstrated only a few atoms (Q3), single devices (Q2) and specific device geometries (Q1).

However, I do genuinely believe and agree that this platform has the potential to overcome all these challenges (although perhaps not demonstrated here), and therefore this manuscript is of value to the community. What a reader would appreciate are descriptions of clear extensions of the current work that would lead to the desired goals (eg., how strong is the atom-photon coupling at 300nm from the surface and how does this compare to other platforms? how to scale up the coupling to multiple devices and atom arrays in one

and two dimensions? what specific challenges would need to be overcome to multiplex atoms across devices? etc)

Q5) What limits the authors from coupling the atoms to the nanophotonic cavity when they achieve a coupling of 20% to the device?

Q6) In Figure 3b, do I understand correctly that ideally the blue datapoints would have no spread, i.e., have the same Stark shift? And if so, the variations are attributed to a combination of several factors such as tweezer power, aberrations, and angle mismatches? Given the high degree of variations across devices, I would have expected a deeper investigation of the main causes, resulting in identification of ways to mitigate them, without which I don't see what additional information 3b provides compared to 3a.

Q7) I found the structure of Figure 3 a bit hard to follow. Fig 3b is hard to understand without understanding Fig 3d and Fig 3c first.

Reviewer #3 (Remarks to the Author):

Reviewer #4 (Remarks to the Author):

This experimental work addresses the important topic of optically connecting neutral atoms in scalable arrays. The ground work in this area for nanofabricated chips has been set by for example Refs. 33, 34, 36, 38, where single-atom manipulation, coupling and movement of atoms near chip-based structures, including tweezer-based control, has been developed.

The results in the manuscript involve scattering light off the atoms in free space above the

chip and gathering information about the atomic location via a spatially-resolved camera using a two-photon excitation. Further, the authors show that atoms are loaded into intensity maxima traps very close to the surface of nanophotonic devices. In each case an array of 4 to 9 atoms controlled via an AOD is used for the demonstration.

The concept I find missing from the paper is quantum optical connection to the atoms through the nanophotonic structure. Certainly free-space imaging is useful for characterization stages of the apparatus. Stark shifts are a good way to characterize the potentials, and atom rearrangement is facilitated by the fluorescence signals observed from the top of the chip. However, in my understanding the ultimate goal of this platform is to enable atom-photon interactions, through which atom detection is also given. The authors could mention in the outlook the needed steps for full operation of a system with an operating photonic interface.

Regarding the background free imaging, a strength of this work is that this is the first time this level scheme has been used for fluorescence imaging of single neutral atoms (in my understanding), although it has been used for ions at the single particle level. Scattering can be a problem in a wide range of cold atom experiments due to windows or other surfaces.

The lifetime information for atoms on the devices is reported in the supplement. I would see this however as a key piece of information. The lifetime on the device is reported to be significantly smaller than free space. While this is not inconsistent with observations in other similar platforms, it would still be worthwhile to comment on the expected contributions to the finite lifetime, and whether in a protocol involving shuttled atoms what the lifetime goals are.

Overall, this manuscript has a number of unique parts and is worthy of publication in Nature Communications.

Reply to Referees

We would like to thank all referees for their thorough evaluation of our manuscript. We very much value the questions and suggestions raised by the referees and provide a point-to-point response below in blue with relevant revisions to the manuscript mentioned in red. Their input has helped us improve the manuscript and further increase its quality and accessibility. Please find the revised manuscript with the changes highlighted attached.

With very best wishes,
Hannes Bernien

Reviewer #1 (Remarks to the Author):

Coupling neutral atoms trapped in optical tweezers with nanophotonic devices features broad applications in quantum information processing and quantum simulation. One crucial technical difficulty in its realization is that standard fluorescence imaging will fail due to strong scattering near the photonic devices. In this article, the authors presented a background-free imaging method by driving a two photon transition to an excited state and imaging one of the decay pathways. With the ability to perform high fidelity imaging, the authors further demonstrated the rearrangement of stochastic loaded optical tweezers into a (small) defect free array, and moving the atoms over the nanophotonic devices. The high fidelity imaging through the background free scheme is the main technical achievement of this paper. The rearrangement and transport of atoms is currently limited to small array sizes (3 - 4 atoms), but the authors are upfront about this and outlines directions for future improvements. In our view there are sufficient technical advances demonstrated here to justify publication in Nature Communications. The manuscript is very clearly written. The data is rigorously analyzed and well-presented. We have a few comments and questions for the authors.

We thank the reviewer for their careful reading and evaluation of our manuscript. We are excited that the reviewer finds the manuscript to be worthy of publication.

1. The paper uses a mix of single shot images and averaged images. These are clearly distinguished in the captions and the main text, but it'll be easier for the reader if they are labeled directly on the images.

Thank you for the suggestion that should help contextualize the images for the readership.

Revision: We have added these labels to the images as recommended by the reviewer.

2. The authors report an imaging fidelity of 99.2%. This number should be put into context with state of the art imaging fidelity for free space optical tweezers, as well as those coupled to photonic crystals.

Our imaging fidelity of 99.2% in free space is comparable to typical data taken using standard fluorescence imaging [8]. Our imaging fidelity of 86% while in the standing wave traps in close proximity to the cavities is slightly lower than the best demonstration to date of ~95% from [50]. Our loss of fidelity is primarily driven by atom loss from the trap which would be improved by lowering atomic temperatures via degenerate Raman sideband cooling as performed in [50]. Our imaging is also performed ~3x faster, further contributing to heating induced atom loss from the trap.

Revision: We have added this context to the main text and supplementary information where these fidelities are discussed.

3. The authors mention that the standing wave trap has ~2x the intensity as a free space tweezer. I would naively expect the field to be 2x higher and intensity to be 4x higher. Could the authors explain this?

We show numerical simulations of the Stark shifts, which are proportional to the trap intensities, for both a free-space tweezer and for the standing wave trap side-by-side in Supplementary Fig. S2 where the peak value of each colorbar shows the ~2x intensity difference. The reason that it is not 4x as one may expect is due to the fact that the beam only partially reflects from the cavity and some trapping intensity is lost to the transmitted beam. We mention this partial reflection in the paper when we describe as follows: "...the tweezer beams partially reflect backward, forming standing wave traps above the devices as shown in Fig. 3d."

Revision: We have further emphasized the partial reflection by editing the caption of Figure 3.

4. The term "magic" (wavelength) is used without explanation. The concept may not be familiar to all readers.

We thank the reviewer for catching this and agree that this term should be defined in text.

Revision: We have added a clause to the paper defining the term to improve comprehension.

5. Legends in Fig 3c are too small.

Thank you for pointing out this issue.

Revision: We have increased the size of the legend accordingly.

6. In image 2 of Fig. 4b, the individual atoms are a bit hard to see by eye. Consider highlighting their locations.

We thank the reviewer for the helpful observation.

Revision: We have added boxes around each atom in the single-shot images of Figure 4 to mark their positions more clearly as suggested by the reviewer.

Reviewer #2 (Remarks to the Author):

The manuscript by Menon*, Glachman* et al reports on an experimental platform that seeks to couple atom arrays to nanophotonic chips. Specifically, they address two challenges: (1) loading atoms near chip-scale structures while retaining the reconfigurability and control of free-space tweezers and (2) imaging the atoms with high fidelity when they are in the vicinity of the chip structures. They tackle the first challenge by using tweezer arrays to stochastically load atoms from a MOT, rearranging the tweezers to achieve unity filling, and moving the tweezers to the chips using radiofrequency control of the AODs. They tackle the second challenge by spectrally removing the exciting wavelengths and filtering only the 895nm fluorescence, achieving background-free imaging. Integrating individual atoms with nanophotonic structures is an active field of research with implications for networked quantum systems for quantum computation and communication.

The paper is well-written and concise, and the claims are well supported by evidence. Prior work is appropriately acknowledged, and the methods are described in sufficient detail to ensure reproducibility of the results.

We thank the reviewer for their consideration of our manuscript and their positive comments about our work.

While I believe the technical advances presented here are significant for the field, the scale and novelty of the advances need to be justified in the context of other

works which have addressed and overcome similar challenges of atom loading (refs 43-49) and imaging (refs 41,48,50) near surfaces.

Below are questions that need to be clarified before I can recommend publication in Nature Communications.

Q1) Claims on generality of chip geometry: In page 1, the authors claim previous attempts have “been limited to... specific device geometries” and imply that their work addresses the outstanding challenge of “readout of large atoms arrays near arbitrary nanophotonic devices”. Later, they claim “the techniques... represent a general recipe”.

While the semi-open ship design is indeed very clever, I don't see how such claims of generality are justified when the presented designs are simply well-established 1D cavity waveguides. How would arbitrary 2D nanophotonic devices be integrated with atom arrays with the presented design? In the Discussion, the authors briefly mention extensions to “loading tracks fabricated on the chip” which could potentially lead to general recipes with arbitrary structures, but without providing further clarity on how such an extension is feasible, one might as well say that their design simply represents another “specific device geometry”.

We thank the reviewer for the thorough comment. The statement “rearrangement and single-shot readout of large atom arrays near arbitrary nanophotonic devices remains an outstanding challenge.” is in the context of imaging atoms near nanophotonic structures. The imaging scheme presented here is agnostic to the number of atoms, underlying device geometry, smoothness, angles, or the distance to the nearby solid surface. This is in contrast to other works where one or more of these conditions have to be met. However, as the reviewer rightly points out, the fluorescence data presented in this work is limited to 1D photonic structures that we have fabricated.

The cavity/waveguide designs presented in this work are 1D by choice due to the highest atom-cavity interaction possible due to extremely small mode volumes [41]. Loading to 2D structures is not explored in this work. However, this platform allows for atoms to be coupled to multiple devices at the same time, a capability we believe is not demonstrated elsewhere. The platform also allows combining different 1D structures such as alligator waveguides [38] where atoms are trapped in between the devices, corrugated cavities where atoms are proposed to couple to the cavity side [57], combined 1D cavities [58] along with the kind of cavities presented in this work.

Revision: We have modified the introduction statement to read that this platform forms a general recipe to combine and load atoms to a wide range of photonic devices mentioned above.

Q2) How do the authors propose to work with multiple nanophotonic devices (eg., scenario in 4b) where light would have to be coupled with good mode-matching into the tapered ends of each device? The present work only demonstrates coupling to a single device using a focussed free-space beam from outside the chamber.

For instances where the cavities or waveguides are read out in a serial manner, an external control on the position of the coupling lens can be used to deterministically couple to the device of choice. For coupling to multiple devices in a parallel fashion, acousto-optical deflectors (AODs) [supplementary reference 8] or fiber arrays can be used.

Revision: We have added a brief discussion of some of these potential methods to the free-space coupling part of the Supplementary Information.

Q3) In the Discussion, the authors say “we load large arrays of atoms near the nanophotonic chip”, which seems unwarranted given that atom array experiments these days (including work by this group) routinely work with hundreds of atoms. The atom numbers presented here ($2 < n < 10$) are comparable to similar works of atoms near nanophotonic surfaces (refs 45, 48).

In Fig. 2, we show an 8x8 array of tweezers trapping atoms within microns of our nanophotonics which to our knowledge is the largest number of atoms with individual tweezer positional control loaded within millimeters of a nanophotonic structure. The size of this array is currently limited by our optical tweezer laser power. The ability to engineer interactions between a part of this array and photonic degrees of freedom is one of the enabling capabilities for distributed quantum computing and clock networks [61]. For 8x8 arrays, we have used a combination of AWG and RF oscillators, however, for the faster readout and rearrangement used in Fig. 4, RF oscillators must be used. For loading atoms onto the devices, we are limited to nine or fewer tweezers due to the limited number of RF tones our first-generation Quantum Machines OPX can provide to drive our AODs. These technical limitations are not fundamental to the methods and techniques that were developed and demonstrated in this work.

Q4) Novelty of advances: The authors cite previous works that have tackled the challenges of loading (refs 43-49) and imaging (refs 41,48,50) atoms near nanophotonic surfaces. Following this, they state “However, these techniques have been limited to either a few atoms, single devices, or specific device geometries”, implying that their overcoming of all these challenges in one platform is the novelty. I would argue that the results in manuscript also demonstrated only a few atoms (Q3), single devices (Q2) and specific device geometries (Q1). However, I do genuinely believe and agree that this platform has the potential to overcome all these challenges (although perhaps not demonstrated here), and

therefore this manuscript is of value to the community. What a reader would appreciate are descriptions of clear extensions of the current work that would lead to the desired goals (eg., how strong is the atom-photon coupling at 300nm from the surface and how does this compare to other platforms? how to scale up the coupling to multiple devices and atom arrays in one and two dimensions? what specific challenges would need to be overcome to multiplex atoms across devices? etc)

We thank the reviewer for believing that our platform has the potential to overcome the challenges raised and hope our answers to questions Q1-Q3 address those concerns. The protocol we are working towards implementing is detailed in a previous work [59] where we have estimated the atom-photon coupling strength at our trap distance to be $\sim 2\pi * 600$ MHz, comparable to values observed in similar experiments [41]. These values are two orders of magnitude larger than what is typically possible with mirror cavities [63] and an order of magnitude larger than what are reported for fiber cavities [31] due to the small mode volumes of 1D photonic cavities. Further, the platform presented here shows that multiple such cavities can be combined in a single system. The additions made to the draft to address Q1-Q3 and Q5 hopefully address the next steps required and challenges related to multiplexed operation of atom arrays and cavity arrays.

Revision: We have added the expected atom-cavity coupling strength at the trapping region along with how it compares to the other platforms.

Q5) What limits the authors from coupling the atoms to the nanophotonic cavity when they achieve a coupling of 20% to the device?

The cavities were not resonant with the atomic transition in this work. For future experiments we plan to incorporate the ability to tune the device resonances thermally, however, this is out of scope for the current manuscript.

Revision: We have modified the discussion section to clarify that the device is not on resonance and that closer fabrication tolerance and thermal tuning can be incorporated in future iterations to overcome this challenge.

Q6) In Figure 3b, do I understand correctly that ideally the blue datapoints would have no spread, i.e., have the same Stark shift? And if so, the variations are attributed to a combination of several factors such as tweezer power, aberrations, and angle mismatches? Given the high degree of variations across devices, I would have expected a deeper investigation of the main causes, resulting in identification of ways to mitigate them, without which I don't see what additional information 3b provides compared to 3a.

Thank you for this careful observation. Fig. 3a shows that atoms are trapped above the devices but provides no information about the distance between the atom and the device. Particularly, multiple intensity maxima can be present within the Rayleigh range of the objective (as shown in Supplementary Fig. S2) with varying distances from the device. Fig. 3b is required to quantify this distance from the device and to understand the occupation of different intensity maxima. Indeed the blue datapoints would have no spread ideally. We agree that many parameters affect this, in particular, we have identified a small relative angle between the chip and the focal plane of the tweezers. This angle combined with the sensitivity of the loading to different intensity maximas with respect to the relative Z positions of the tweezer focus and the nanophotonic leads to an expected large variation. Correcting the angle is possible by changing the angle of the objective to reduce this variation. However, this introduces coma aberration on the tweezers as the objective is corrected for glass thickness only at one angle. For future experiments, we can overcome this by adjusting the chip angle prior to mounting or by using a rotatable chip mount. However, these are outside the scope of this work.

Revision: We have adjusted the wording in the discussion where we mention the angle between the devices and the tweezer focal plane to clarify that this angle strongly affects the loading distribution to the various intensity maxima across devices.

Q7) I found the structure of Figure 3 a bit hard to follow. Fig 3b is hard to understand without understanding Fig 3d and Fig 3c first.

We thank the referee for pointing this out.

Revision: We have reordered the subfigures and adjusted the text accordingly to have a more logical flow of the story.

Reviewer #3 (Remarks to the Author):

We thank Reviewer #3 for participating in the co-peer review initiative.

Reviewer #4 (Remarks to the Author):

This experimental work addresses the important topic of optically connecting neutral atoms in scalable arrays. The ground work in this area for nanofabricated chips has been set by for example Refs. 33, 34, 36, 38, where single-atom manipulation, coupling and movement of atoms near chip-based structures, including tweezer-based control, has been developed.

The results in the manuscript involve scattering light off the atoms in free space above the chip and gathering information about the atomic location via a spatially-resolved camera using a two-photon excitation. Further, the authors show that atoms are loaded into intensity maxima traps very close to the surface of nanophotonic devices. In each case an array of 4 to 9 atoms controlled via an AOD is used for the demonstration.

The concept I find missing from the paper is quantum optical connection to the atoms through the nanophotonic structure. Certainly free-space imaging is useful for characterization stages of the apparatus. Stark shifts are a good way to characterize the potentials, and atom rearrangement is facilitated by the fluorescence signals observed from the top of the chip. However, in my understanding the ultimate goal of this platform is to enable atom-photon interactions, through which atom detection is also given. The authors could mention in the outlook the needed steps for full operation of a system with an operating photonic interface.

We thank the reviewer for carefully going through the manuscript and the comments. Imaging indeed is a great tool to characterize the atomic position with respect to the chip. Imaging is also useful beyond characterization, as it provides access to parallel state readout both in free space and while coupled to devices. For atom-array based computation/simulation in the presence of photonic chips, this becomes an essential readout mechanism. Further, when multiple atoms are coupled to a single device, the photons collected through the nanophotonic interaction do not provide atom-by-atom state information. In these cases, the imaging scheme allows for the site-dependent atomic state resolution.

As the reviewer rightly points out, our goal with this platform is indeed to demonstrate a fully functional atom-photon interface as discussed in our previous proposal [59]. This requires the cavities to be on resonance with the atomic transition and involves the capability to thermally tune and stabilize the resonance

of the cavity. In future iterations of the experiments, we will be working to incorporate this capability.

Revision: We have modified the discussion section to include the technical requirements to generate light-matter interaction.

Regarding the background free imaging, a strength of this work is that this is the first time this level scheme has been used for fluorescence imaging of single neutral atoms (in my understanding), although it has been used for ions at the single particle level. Scattering can be a problem in a wide range of cold atom experiments due to windows or other surfaces.

We thank the reviewer for this comment. We agree and are also excited that this technique can be applied to a wide range of cold atom experiments limited by scattering challenges.

The lifetime information for atoms on the devices is reported in the supplement. I would see this however as a key piece of information. The lifetime on the device is reported to be significantly smaller than free space. While this is not inconsistent with observations in other similar platforms, it would still be worthwhile to comment on the expected contributions to the finite lifetime, and whether in a protocol involving shuttled atoms what the lifetime goals are.

The lifetime of atoms trapped on the devices is indeed smaller than that of atoms in free space. This is consistent with prior observations for atoms trapped on nanophotonic devices [41] and is understood to be a result of thermally populated phononic modes [supplementary citation 4]. However, the lifetime of atoms on devices presented in this work is more than two orders of magnitude larger than the observed coherence time of atoms trapped on the nanophotonic devices ~ 2 ms [42], four orders of magnitude larger than atom shuttling time ~ 100 us, and more than six orders of magnitude larger than the required atom-cavity interaction timescale for our atom-cavity protocol ~ 100 ns [59]. Due to these short time scales, we believe that the observed lifetime of ~ 700 ms is sufficiently long to have minimal effects on future experiments.

Revision: We have added the relevant discussion and timescales in the supplementary section to clarify the impact of the observed lifetime of atoms trapped on the nanophotonic devices.

Overall, this manuscript has a number of unique parts and is worthy of publication in Nature Communications.

We thank the reviewer for their assessment of our manuscript and recommendation of publication.

REVIEWERS' COMMENTS

Reviewer #1 (Remarks to the Author):

In the revised manuscript, the authors have addressed all my questions and concerns (and, in my view, those raised by the other two reviewers). I recommend publication without further revisions.

Reviewer #2 (Remarks to the Author):

The authors have satisfactorily responded to my questions and made appropriate changes. Therefore I recommend publication of the revised manuscript.

Reviewer #3 (Remarks to the Author):

Reviewer #4 (Remarks to the Author):

The authors have addressed all of my comments and I recommend publication.